# Towards Long-Tailed 3D Detection

Neehar Peri[1], Achal Dave[1], Deva Ramanan[1,2]*, Shu Kong[3]*
[1]Robotics Institute, Carnegie Mellon University
[2]Argo AI, [3]Dept. of Computer Science and Engineering, Texas A&M University
{nperi, adave, deva}@andrew.cmu.edu  shu@tamu.edu

**Abstract:** Contemporary autonomous vehicle (AV) benchmarks have advanced techniques for training 3D detectors, particularly on large-scale LiDAR data. Surprisingly, although semantic class labels naturally follow a long-tailed distribution, these benchmarks focus on only a few `common` classes (e.g., `pedestrian` and `car`) and neglect many `rare` classes in-the-tail (e.g., `debris` and `stroller`). However, in the real open world, AVs must still detect `rare` classes to ensure safe operation. Moreover, semantic classes are often organized within a hierarchy, e.g., tail classes such as `child` and `construction-worker` are arguably subclasses of `pedestrian`. However, such hierarchical relationships are often ignored, which may yield misleading estimates of performance and missed opportunities for algorithmic innovation. We address these challenges by formally studying the problem of *Long-Tailed 3D Detection* (LT3D), which evaluates on *all* classes, including those in-the-tail. We evaluate and innovate upon popular 3D detectors, such as CenterPoint and PointPillars, adapting them for LT3D. We develop hierarchical losses that promote feature sharing across common-vs-rare classes, as well as improved detection metrics that award partial credit to "reasonable" mistakes respecting the hierarchy (e.g., mistaking a `child` for an `adult`). Finally, we point out that fine-grained tail class accuracy is particularly improved via *multimodal fusion* of RGB images with LiDAR; simply put, fine-grained classes are challenging to identify from sparse (LiDAR) geometry alone, suggesting that multimodal cues are crucial to long-tailed 3D detection. Our modifications improve accuracy by 5% AP on average for all classes, and dramatically improve AP for `rare` classes (e.g., `stroller` AP improves from 0.1 to 31.6)!

**Keywords:** Autonomous Vehicles, Long-Tailed 3D Detection, Multimodal Fusion

## 1   Introduction

3D object detection is a key component in many robotics systems such as autonomous vehicles (AVs) [1, 2]. To facilitate research in this space, the AV industry has released large-scale 3D annotated multimodal datasets [2, 3, 4]. However, these datasets benchmark on only a few `common` classes such as `pedestrian` and `car`. In the real open world, safe navigation [5, 6] requires AVs to reliably detect `rare`-class objects such as `child` and `stroller`. This motivates *Long-Tailed 3D Detection* (LT3D), a problem requiring detecting objects from both `common` and `rare` classes.

**Status Quo**. Among contemporary AV datasets, nuScenes [2] has exhaustively annotated objects of various classes crucial to AVs (Fig. 1) and organized them with a semantic hierarchy (Fig. 3). As it focuses on only a few (`common`) classes, prior works miss opportunities to exploit this semantic hierarchy during training. We argue that these benchmarking protocols are flawed because detecting fine-grained classes is useful for downstream tasks such as motion planning. This motivates us to study LT3D by re-purposing *all* annotated classes in nuScenes.

**Protocol**. LT3D requires 3D localization and recognition of objects from each of the `common` (e.g., `adult` and `car`) and `rare` classes (e.g, `child` and `stroller`). Moreover, for safety-critical robots such as autonomous vehicles, we believe detecting but mis-classifying `rare` objects (e.g., mis-classifying a `child` as an `adult`) is preferable to failing to detect them at all. Therefore, we propose a new metric to quantify the severity of classification mistakes in LT3D that exploits inter-class relationships to award partial credit (Fig. 3). We use both the standard and proposed metrics to evaluate 3D detectors on all classes.

6th Conference on Robot Learning (CoRL 2022), Auckland, New Zealand.

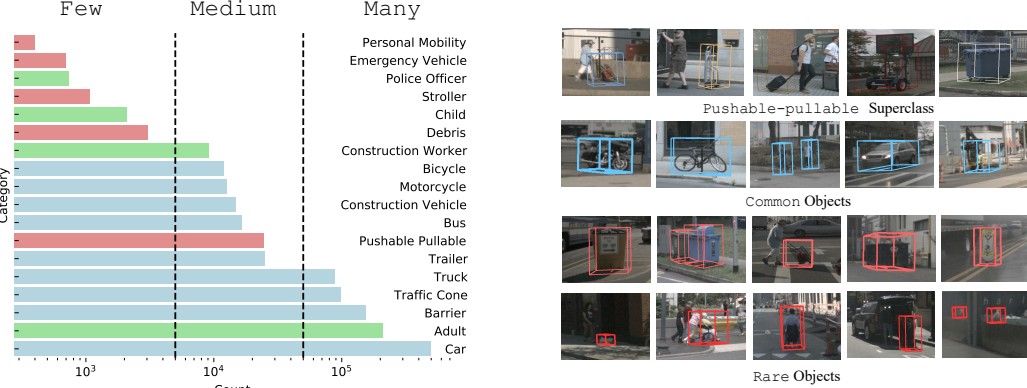

Figure 1: According to the histogram of per-class object counts (on the **left**), the nuScenes benchmark focuses on the common classes in cyan (e.g., `car` and `barrier`) but ignores rare ones in red (e.g., `stroller` and `debris`). In fact, the benchmark creates a superclass `pedestrian` by grouping multiple classes in green, including the common class `adult` and several rare classes (e.g., `child` and `police-officer`); this complicates the analysis of detection performance as `pedestrian` performance is dominated by `adult`. Moreover, the ignored superclass `pushable-pullable` also contains diverse objects such as `shopping-cart`, `dolly`, `luggage` and `trash-can` as shown in the top row (on the **right**). We argue that AVs should also detect `rare` classes as they can affect AV behaviors. Following [7], we report performance for three groups of classes based on their cardinality (split by dotted lines): `Many`, `Medium`, and `Few`.

**Technical Insights**. To address LT3D, we first retrain state-of-the-art LiDAR-based 3D detectors on *all* classes. Naively retraining detectors produces poor performance on `rare` classes (e.g., yielding 0.1 AP on `child` and 0.1 AP on `stroller`). We propose several algorithmic innovations to improve these results. First, to encourage feature sharing across common-vs-rare classes, we learn a single feature trunk, adding in hierarchical coarse classes that ensure features will be useful for both `common` and `rare` classes. Second, we find that LiDAR data is simply too impoverished for even humans to recognize certain tail objects that tend to be small, such as `strollers`. We explore multimodal fusion, and introduce a simple approach that post-processes single-modal 3D detections from LiDAR and RGB inputs, filtering away detections that are inconsistent across modalities. Our innovations significantly improve performance on LT3D by 5 % AP on average, greatly boosting performance when allowing for partial credit (e.g., achieving 16.9 / 38.8 AP for `child` / `stroller`).

**Contributions**. We make three major contributions. First, we formulate the problem LT3D, emphasizing detection of both `common` and `rare` classes in safety-critical AVs. Second, we design LT3D's benchmarking protocol and develop a supplemental metric that awards partial credit depending on the severity of misclassifications (e.g., misclassifying `child-vs-adult` is less problematic than misclassifying `child-vs-car`). Third, we propose several architecture-agnostic approaches to LT3D, including a simple multimodal fusion technique that uses RGB-based detections to filter out false-positive LiDAR-based detections, leading to significant improvement of `rare`-class detection.

## 2 Related Works

**3D Object Detection**. Contemporary benchmarks, often in the AV setting, favor LiDAR-based detectors, emphasizing `common` classes and ignoring `rare` ones. Approaches to 3D detection usually adopt an anchor-based model architecture that defines per-class shapes to guide class-aware object detection [8, 9, 10, 11, 12]. A recent *anchor-free* model, CenterPoint [13] achieves the state-of-the-art for LiDAR-based 3D object detection. Specifically, it learns to predict an object's center and estimates the 3D shape for each detected object's center. Existing LiDAR-based 3D detectors exclusively focus on data from `common` classes [8, 9, 13] and do not study how to detect `rare` classes. RGB-based 3D detectors underperform LiDAR-based methods because the monocular RGB input does not provide reliable 3D measures (unlike LiDAR). As a result, RGB-based 3D detectors are not widely adopted. However, in exploring LT3D we find that RGB-detectors shine for detecting objects of `rare` classes. Importantly, multimodal fusion significantly improves LT3D.

**Multimodal 3D Detection**. Conventional wisdom suggests that fusing multimodal cues, particularly using LiDAR and RGB, can improve 3D detection. Intuitively, LiDAR faithfully measures the 3D world (although it has notoriously sparse point returns), and RGB is high-resolution (but lacks 3D information). Multimodal fusion for 3D detection is an active field of exploration. Existing methods suggest different ways to fuse the two modalities. Proposed methods encode separate modalities and fuse object proposals [14, 15, 16, 17, 18, 19], augment LiDAR points with either RGB features [20], augment RGB images with LiDAR points [21] or add semantic information obtained by processing RGB inputs [22, 23]. Others propose stage-wise methods that first detect boxes over images and localize in 3D with LiDAR [24] and fuse detections from single-modal detectors [25, 26]. While the above methods have not been tested for LT3D, our work shows that RGB is a key modality for LT3D.

**Long-Tailed Perception** (LTP). Real-world data tends to follow long-tailed class distributions [27], i.e., a few classes are dominant in the data, while many others are rarely seen. LTP is a long-standing problem in the literature [7]. It has been widely studied through the lens of image classification and requires training on class-imbalanced data, aiming for high accuracy averaged across imbalanced classes [7, 28, 29]. Existing methods propose reweighting losses [30, 31, 32, 33, 34, 35], rebalancing data sampling [36, 37, 38], balancing gradients computed from imbalanced classes [39], and balancing network weights [29]. Others study LTP through the lens of 2D object detection over RGB images [40]. Compared to 2D image-based recognition, 3D long-tailed detection has unique opportunities and challenges because sensors such as LiDAR directly provide geometric and ego-motion cues that are difficult to extract from 2D images. 2D detectors must detect objects of different scales due to perspective image projection, dramatically increasing the complexity of the output space (e.g., requiring more anchor boxes). In contrast, 3D objects do not exhibit as much scale variation, but far-away objects tend to be sparse, imposing different challenges. Finally, 3D detectors often use class-aware heads (i.e. each class has its own binary classifier) while 2D long-tail recognition typically use shared softmax heads (which may make it easier to enforce hierarchies, as explained above). To the best of our knowledge, long-tailed 3D detection (LT3D) has not yet been explored. In LT3D, we find a unique challenge: rare classes are not only infrequent but are also difficult to distinguish using LiDAR alone. This motivates us to use RGB to complement LiDAR. We find using both RGB (for better recognition) and LiDAR (for better 3D localization) helps detect `rare` classes.

## 3 LT3D: Methodology

To approach LT3D, we first retrain SOTA 3D detectors on *all* classes, including LiDAR-based detectors (PointPillars [8]nand CenterPoint [13]), an RGB-based detector (FCOS3D [41]), and multimodal detector (TransFusion [17]). We further introduce several modifications that consistently improve their LT3D performance. Please refer to the supplement for training details.

**Grouping-Free Detector Head**. Extending existing 3D detectors to train with more classes is surprisingly challenging. Many contemporary networks use a multi-head architecture that groups classes of similar size and shape to facilitate efficient feature sharing. For example, CenterPoint groups `pedestrian` and `traffic-cone` since these objects are both tall and skinny. However, multi-headed grouping strategies may not work for diverse classes like `pushable-pullable` and `debris` and are difficult to scale for a large number of classes. Therefore, we first consider making each class its own group to avoid hand-crafted grouping heuristics. However, the multi-head architecture has per-class detectors that consist of multiple layers with lots of parameters, hence learning them easily overfits to rare-classes. Our final solution is to merge all classes into a single group with a proportionally heavier detector head to simplify training. Our group-free (i.e. single-head) architecture has a shared backbone across all classes, and each class has only one linear layer as its detector. This significantly reduces the number of parameters and allows learning the shared feature backbone collaboratively with all classes, effectively mitigating overfitting to rare-classes. Adding a new class is as simple as adding a single linear layer to the detector head. Our grouping-free detector head achieves improved accuracy over grouping-based methods, as shown in the supplement.

**Training with Semantic Hierarchies**. nuScenes defines a semantic hierarchy (Fig. 3) for all classes, grouping semantically similar classes under coarse-grained categories. We leverage this hierarchy during training. Specifically, we train detectors to predict three labels for each object: its fine-grained label (e.g., `child`, its coarse class (e.g., `pedestrian`), and the root class `object`. We adopt a grouping-free detector head that outputs separate "multitask" heatmaps for each class, and use a per-class sigmoid focal loss rather than multi-class cross-entropy loss. It is worth noting that this

| LiDAR-based Detections | RGB-based Detections | Filtered LiDAR-based Detections |
|:---:|:---:|:---:|
| 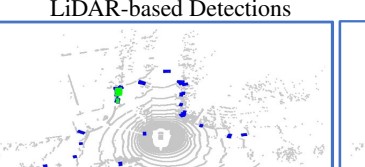 | 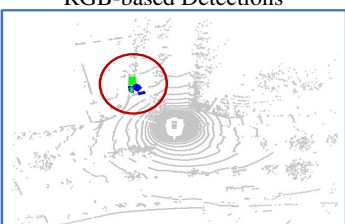 | 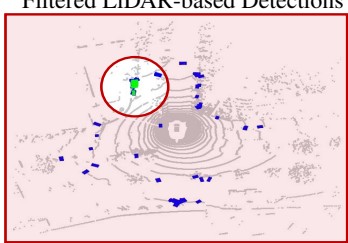 |

Figure 2: Multimodal filtering effectively removes high-scoring false-positive LiDAR detections. The green boxes are ground-truth `strollers`, while the blue boxes are `stroller` detections from our best performing models, liDAR-based detector CenterPoint [13] (**left**) and RGB-based detector FCOS3D [41] (**mid**). The final filtered result removes LiDAR detections not within $m$ meters of any RGB detection (**right**).

simple "multitask" learning strategy does not necessarily enforce a hierarchy, hence can extend to more complex label relationships. Crucially, because we do not employ softmax losses, adding a `vehicle` heatmap does not directly interfere with the `car` heatmap (as they would with a multi-class softmax loss). However, this might produce repeated detections on the same test object. We address that by simply ignoring coarse detections at test time. We explore alternatives in the supplement and conclude that they achieve similar LT3D performance. Perhaps surprisingly, this training method improves detection performance not only for `rare` classes, but also for `common` classes.

**Augmentation Schedule**. Class-balanced resampling is a common technique in learning with long-tailed classes. This augmentation strategy increases the number of `rare` objects seen in training but skews the class distribution and leads to more false positives for `rare` classes in inference. Prior works [22, 17] suggest disabling class-balanced resampling for the last few training epochs to better match the real class distribution, reducing false positives. We validate this approach in training 3D detectors and find that it often improves performance for `rare` classes at the cost of `common` classes.

**Multimodal Fusion by Filtering**. Small fine-grained classes are challenging to identify from sparse (LiDAR) geometry alone, suggesting that multimodal cues can improve long-tailed detection. We evaluate several multimodal fusion algorithms, but find a simple strategy of post-hoc filtering to work remarkably well. Although LiDAR-based detectors are widely adopted for 3D detection, we find that they produce many high-scoring false positives (FPs) for rare classes due to misclassification. We focus on removing such FPs. To this end, we use an RGB-based detector to filter out high-scoring false-positive LiDAR detections by leveraging two insights: (1) LiDAR-based 3D-detectors are accurate w.r.t 3D localization and yield high recall (though classification is poor), and (2) RGB-based 3D-detections are accurate w.r.t recognition (though 3D localization is poor). Fig. 2 demonstrates this filtering strategy. For each RGB-based detection, we keep LiDAR-based detections within a radius of $m$ meters and remove all the others (that are not close to any RGB-based detections). We use FCOS3D [41] as the RGB-based detector in this work.

## 4   LT3D: Evaluation Protocol

Conceptually, LT3D extends the traditional 3D detection problem, which focuses on identifying objects from $K$ `common` classes, by further requiring detection of $N$ `rare` classes. As LT3D emphasizes detection performance on *all* classes, we report the metrics for three groups of classes based on their cardinality (Fig. 1-left): *many* ($>$50k instances per class), *medium* (5k$\sim$50k), and *few* ($<$5k). We describe the metrics below.

**Standard Detection Metrics**. Mean average precision (mAP) is an established metric for object detection [42, 1, 43]. For 3D detection on LiDAR sweeps, a true positive (TP) is defined as a detection that has a center distance within a distance threshold on the ground-plane to a ground-truth annotation [2]. mAP computes the mean of AP over classes, where per-class AP is the area under the precision-recall curve, and distance thresholds of [0.5, 1, 2, 4] meters.

**Hierarchical Mean Average Precision (mAP$_H$)**. For safety critical applications, although correctly localizing and classifying an object is ideal, detecting but misclassifying *some* object is more desirable than a missed detection (e.g., detect but misclassify a `child` as `adult` versus not detecting this `child`). Therefore, we introduce hierarchical AP (AP$_H$) which considers such semantic relationships

across classes to award partial credit. To encode these relationships between classes, we leverage the semantic hierarchy (Fig. 3) defined by nuScenes. We derive partial credit as a function of semantic similarity using the least common ancestor (LCA) distance metric. Hierarchical metrics have been proposed for image classification [44], but have not been explored for object detection. Extending this metric for object detection is challenging because we must consider how to jointly evaluate semantic and spatial overlap. For clarity, we will describe the procedure in context of computing $AP_H$ for some arbitrary class $C$.

**LCA=0**: Consider the predictions and ground-truth boxes for $C$. Label the set of predictions that overlap with ground-truth boxes for $C$ as true positives. Other predictions are false positives. *This is identical to the standard AP metric.*

**LCA=1**: Consider the predictions for $C$, and ground-truth boxes for $C$ and all sibling classes of $C$ (that have LCA distance to $C$ of 1). Label the set of predictions that overlap a ground-truth box of $C$ as a true positive. Label the set of predictions that overlap sibling classes as `ignored` [43]. All other predictions for $C$ are false positives.

**LCA=2**: Consider the predictions for $C$ and ground-truth boxes for $C$ and all sibling classes of $C$ (that have LCA distance to $C$ less than 2. For nuScenes, this includes all classes.) Label the set of predictions that overlap ground-truth boxes for $C$ as true positives. Label the set of predictions that overlap other classes as `ignored`. All other predictions for $C$ are false positives.

## 5    Experiments

We conduct experiments to better understand the LT3D problem, and gain insights by validating our techniques described in Section 3. Specifically, we aim to answer the following questions:[1]

1. Are `rare` classes more difficult to detect than `common` classes?
2. Are objects from `rare` classes sufficiently localized but mis-classified?
3. Does training with the semantic hierarchy improve detection performance for LT3D?
4. Does multimodal fusion help detect `rare` classes?

We start this section by introducing the model architecture, implementation and dataset.

**Model Architecture**. For LiDAR-based 3D detectors, we use PointPillars [8] and CenterPoint [13], which are widely benchmarked in the literature. For fusion-based 3D detectors, we use TransFusion [17], a recently released state-of-the-art method. TransFusion proposes an end-to-end DETR-like approach [45] for multimodal 3D detection.

**Implementation**. We use the PyTorch toolbox [46] to train all models for 20 epochs with the Adam optimizer [47] and a one-cycle learning rate scheduler [48]. In training, we adopt data augmentation techniques introduced by [13]. When using the introduced data augmentation schedule (cf. Section 3), we train models for 15 epochs with data augmentation enabled, and 5 epochs without. We further describe our implementation in the supplement.

**Datasets.** We use nuScenes [2] and Argoverse 2.0 (AV2) [49] to explore LT3D. Both have fine-grained classes (18 and 26 classes in nuScenes and AV2 respectively) that follow long-tailed distributions. To quantify the long-tail, we use the imbalance factor (IF) defined as the ratio between the numbers of annotations of the max-class and min-class [32]; nuScenes and AV2 have IF=1670 and 2500 respectively – significantly more imbalanced than existing long-tail image recognition benchmarks, e.g., iNaturalist (IF=500) [50] and ImageNet-LT (IF=1000) [51]. NuScenes arranges classes in a semantic hierarchy (Fig. 3); AV2 does not provide a semantic hierarchy but we construct one based on the nuScenes' hierarchy. Following prior work, we use their official train-sets for training and evaluate on the their official val-sets. We focus on nuScenes in the main paper and AV2 in the supplement. Our primary conclusions hold for both datasets.

**Retraining state-of-the-art 3D detectors for LT3D**. We retrain several 3D detectors, namely FCOS3D [41], PointPillars [8] and CenterPoint [13]. FCOS3D operates on monocular images. The other three detectors take an aggregated stack of ten LiDAR-sweeps as input. All four models predict 3D bounding boxes for 18 classes as defined by the nuScenes LT3D protocol. As shown in

---

[1]Answers: yes, yes, yes, yes.

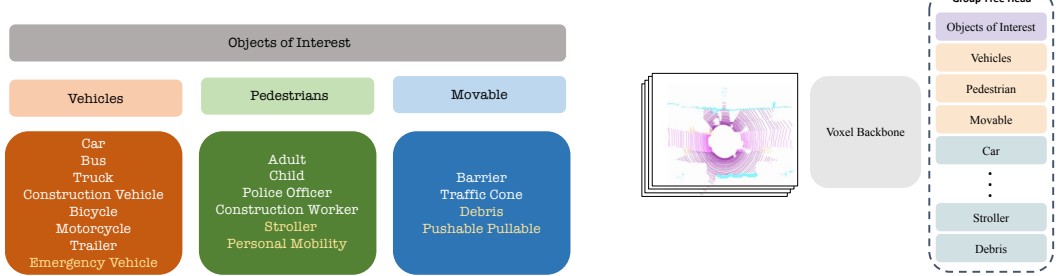

Figure 3: nuScenes defines a semantic hierarchy (on the **left**) for all annotated classes (Fig. 1). We highlight `common` classes in white and `rare` classes in gold. The standard nuScenes benchmark makes two choices for dealing with rare classes: (1) ignore them (e.g., `stroller` and `pushable-pullable`), or (2) group them into coarse-grained classes (e.g., `adult`, `child`, `construction-worker`, `police-officer` are grouped as `pedestrian`). Since the `pedestrian` class is dominated by `adult` (Fig. 1), the standard benchmarking protocol masks the challenge of detecting rare classes like `child` and `police-officer`. We leverage this hierarchy during training (on the **right**) by predicting class labels at *multiple* levels of the hierarchy. Specifically, we train detectors to predict three labels for each object: its fine-grained label (e.g., `child`, its coarse class (e.g., `pedestrian`), and the root-level class `object`. This means that the final vocabulary of classes is no longer mutually exclusive, complicating the application of multi-class softmax losses. To address this, use a sigmoid focal loss that learns separate spatial heatmaps for each class.

Table 1: **Benchmarking detectors for LT3D** (measured by mAP). We present several salient conclusions. First, training with the semantic hierarchy improves all methods for LT3D, e.g., improving by 1% AP averaged over `All` classes. Second, multimodal filtering yields between 4∼11 AP improvement on `Medium` and `Few` classes! This is surprising given that FCOS3D is a less powerful 3D detector on its own. Importantly, we do not modify FCOS3D for LT3D. Interestingly, it also improves multimodal detectors (1.6% AP improvement for TransFusion on `All` classes), demonstrating the importance of using RGB to improve LT3D with better recognition. Third, perhaps surprisingly, post-hoc multimodal filtering of LiDAR-only detector CenterPoint with RGB-only detector FCOS3D performs the best, surpassing the multi-modal TransFusion model. Lastly, data augmentation schedules do not necessarily improve LT3D performance, demonstrating the challenge of 3D detection in the long-tail.

| Method | Multimodal | Many | Medium | Few | All |
|---|---|---|---|---|---|
| FCOS3D (RGB-only) [41] | | 39.0 | 23.3 | 2.9 | 20.9 |
| PointPillars (LiDAR-only) [8] | | 64.2 | 28.4 | 3.4 | 30.0 |
| + Hierarchy | | **66.4** | 30.4 | 2.9 | 31.2 |
|     w/ Data Aug. | | 54.4 | 24.2 | 1.8 | 25.1 |
|     w/ Multimodal Filtering | ✓ | 66.2 | **41.0** | **4.4** | **35.8** |
| CenterPoint (LiDAR-only) [13] | | 76.4 | 43.1 | 3.5 | 39.2 |
| + Hierarchy | | **77.1** | 45.1 | 4.3 | 40.4 |
|     w/ Data Aug. | | 73.8 | 44.5 | 7.4 | 40.3 |
|     w/ Multimodal Filtering | ✓ | **77.1** | **49.0** | **9.4** | **43.6** |
| TransFusion-L (LiDAR-only)[17] | | 68.5 | **42.8** | 8.4 | 38.5 |
|     w/ Multimodal Filtering | ✓ | 73.2 | 42.5 | 8.3 | 39.6 |
| TransFusion (LiDAR + RGB) | ✓ | **73.9** | 41.2 | **9.8** | 39.8 |
|     w/ Data Aug. | ✓ | 73.4 | 40.9 | 8.2 | 39.0 |
|     w/ Multimodal Filtering | ✓ | **73.9** | 42.5 | 9.1 | **40.1** |

Table 1, LiDAR-based detectors that perform well on `common` classes tend to also perform well on `rare` classes.

**Training with Semantic Hierarchy.** Next, we modify our LiDAR-based detectors to jointly predict class labels at different levels of the semantic hierarchy. For example, we modify the detector to additionally classify `stroller` as `pedestrian` and `object`. The semantic hierarchy naturally groups classes based on shared attributes and may have complementary features. Moreover, training with the semantic hierarchy allows `rare` classes within each group to learn better features by sharing with `common` classes. This approach is generally effective, as shown in Table 1, improving accuracy for classes with `Many` examples by 2%, `Medium` examples by 2% and `Few` examples by 1% AP.

Table 2: **Diagnosis using the mAP$_H$ metric on selected classes**. We analyze the best-performing LiDAR-only model CenterPoint and multimodal model TransFusion, with / without our hierarchical loss (*hier.*) and mutimodal filtering technique (*filtering*). Comparing the rows of LCA=0 for with and without our techniques (for CenterPoint and TransFusion respectively), we see our techniques bring significantly improvements on classes with `medium` and `few` examples such as `construction-vehicle` (CV), `bicycle`, `motorcycle` (MC), `construction-worker` (CW), `stroller`, and `pushable-pullable` (PP). Moreover, performance increases significantly from LCA=0 to LCA=1 compared against LCA=1 to LCA=2, (Table 2), confirming that objects from `rare` classes are often detected but misclassified as some sibling classes.

| Method | $mAP_H$ | Car | Adult | Truck | CV | Bicycle | MC | Child | CW | Stroller | PP |
|---|---|---|---|---|---|---|---|---|---|---|---|
| CenterPoint (OTS) | LCA=0 | 82.4 | 81.2 | 49.4 | 19.7 | 33.6 | 48.9 | 0.1 | 14.2 | 0.1 | 21.7 |
| | LCA=1 | 83.9 | 82.0 | 58.7 | 20.5 | 35.2 | 50.5 | 0.1 | 18.3 | 0.1 | 22.0 |
| | LCA=2 | 84.0 | 82.4 | 58.8 | 20.7 | 36.4 | 51.0 | 0.1 | 19.5 | 0.1 | 22.6 |
| CenterPoint (Group-Free) | LCA=0 | 88.1 | 86.3 | 62.7 | 24.5 | 48.5 | 62.8 | 0.1 | 22.2 | 4.3 | 32.7 |
| | LCA=1 | 89.0 | 87.1 | 71.6 | 26.7 | 50.2 | 64.7 | 0.1 | 29.4 | 4.5 | 32.9 |
| | LCA=2 | 89.1 | 87.5 | 71.7 | 26.8 | 51.1 | 65.2 | 0.1 | 30.5 | 4.8 | 33.4 |
| CenterPoint (Group-Free) *w/ Hierarchy* | LCA=0 | **88.6** | **86.9** | 63.4 | 25.7 | 50.2 | 63.2 | 0.1 | 25.3 | 8.7 | 36.8 |
| | LCA=1 | **89.5** | **87.6** | 72.4 | 27.5 | 52.2 | 65.2 | 0.1 | 32.4 | 9.4 | 37.0 |
| | LCA=2 | **89.6** | **88.0** | 72.5 | 27.7 | 53.2 | 65.7 | 0.1 | 34.0 | 9.8 | 37.6 |
| CenterPoint (Group-Free) *w/ Hier. & MM. Filtering* | LCA=0 | 88.5 | 86.6 | **63.4** | **29.0** | **58.5** | **68.2** | **5.3** | **35.8** | **31.6** | **39.3** |
| | LCA=1 | 89.4 | 87.4 | **72.4** | **31.3** | **61.2** | **69.7** | 15.2 | **52.0** | **37.7** | **39.4** |
| | LCA=2 | 89.5 | 87.7 | **72.5** | **31.5** | **62.3** | **69.9** | 16.9 | **56.3** | **38.8** | **39.8** |
| TransFusion-L (LiDAR-only) | LCA=0 | 84.4 | 84.5 | 58.5 | 15.1 | 44.9 | 57.2 | 1.0 | 15.1 | 3.2 | 19.6 |
| | LCA=1 | 85.5 | 85.7 | 67.4 | 21.8 | 46.7 | 59.1 | 1.6 | 21.8 | 3.7 | 19.8 |
| | LCA=2 | 85.5 | 86.1 | 67.5 | 22.6 | 47.7 | 59.9 | 1.7 | 22.6 | 4.2 | 20.4 |
| TransFusion (LiDAR + RGB) | LCA=0 | 84.4 | 84.2 | 58.4 | 24.5 | 46.7 | 60.8 | 3.1 | 21.6 | 13.3 | 25.3 |
| | LCA=1 | 86.0 | 85.4 | 67.3 | 26.3 | 50.1 | 63.5 | 14.4 | 34.7 | 20.6 | 25.6 |
| | LCA=2 | 86.0 | 85.9 | 67.4 | 26.8 | 52.2 | 65.1 | 15.2 | 36.1 | 22.8 | 26.4 |
| TransFusion *w/ Multimodal Filtering* | LCA=0 | 84.4 | 84.2 | 58.4 | 25.3 | 52.3 | 62.8 | 4.0 | 27.5 | 14.7 | 27.3 |
| | LCA=1 | 86.0 | 85.4 | 67.3 | 26.6 | 55.7 | 64.0 | **25.1** | 46.7 | 24.3 | 27.4 |
| | LCA=2 | 86.0 | 85.9 | 67.4 | 27.0 | 56.9 | 64.3 | **25.8** | 48.6 | 28.3 | 27.9 |

**Data Augmentation Schedule**. Prior works [25, 17] suggest disabling copy-paste augmentation for the last few epochs of training to reduce the number of false positive detections. We validate this claim for various detector architectures and find that although it seems to help `rare` classes by 3% AP, but hurts `common` classes by 4% AP (c.f. CenterPoint).

**Multimodal Fusion via Filtering.** Detecting `rare` classes from LiDAR-only is challenging since its difficult to recognize objects from sparse LiDAR points alone. As a result, LiDAR-detectors often have many high-scoring FPs (Fig. 2), resulting in low AP. Using RGB detections to filter the LiDAR detections results in significant performance improvement on `rare` classes, improving by 4∼11 AP on classes with `Medium` and `Few` examples for all models (Table 1)!

**End-to-End Multimodal Methods**. Since multimodal cues significantly improve LT3D, we are motivated to explore end-to-end approaches. Specifically, we retrain TransFusion [17] on all 18 classes. We retrain TransFusion, downsampling the RGB images by a factor of 2 to fit the model in GPU memory. Surprisingly, lidar-only and multi-modal variants of TransFusion perform worse than CenterPoint for LT3D. Further hyperparameter tuning and full-resolution training may help. Lastly, our multimodal filtering strategy still improves this end-to-end fusion methods slightly, e.g., it increases mAP on `All` classes by 0.3% AP for TransFusion (cf. Table 1). We also find that applying multimodal filtering on the LiDAR-only branch of TransFusion yields similar performance to the multimodal model, suggested that TransFusion is simply learning a multimodal filtering function.

**Analysis of Misclassifications**. For 3D detection, localization and classification are two important measures of 3D detection performance. In practice, we cannot achieve perfect performance for either. In safety-critical applications, detecting but misclassifying objects (as a semantically related category) is more desirable than a missed detection (e.g., detect but misclassify a `child` as `adult` versus not detecting this `child`). Therefore, we introduce hierarchical AP (AP$_H$), which considers such semantic relationships across classes to award partial credit. Applying this hierarchical AP reveals

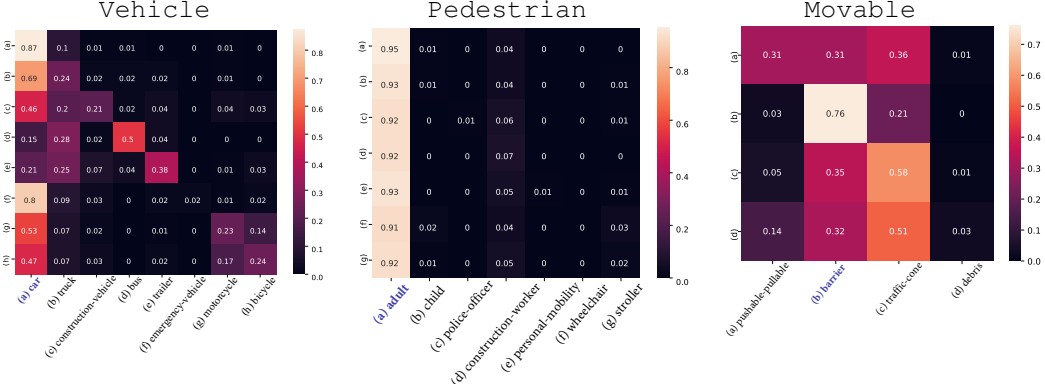

Figure 4: **Breakdown analysis of misclassifications within superclasses.** We analyze our best-performing model (CenterPoint w/ hierarchical training and multimodal filtering). Fine-grained classes are most often confused by the dominant class (in blue) in each superclass: (**left**) `Vehicle` is dominated by `car`, (**mid**) `Pedestrian` is dominated by `adult`, and (**right**) `Movable` is dominated by `barrier`. We find that class confusions are reasonable. `Car` is often mistaken for `truck`. Similarly, `truck`, `construction-vehicle` and `emergency-vehicle` are most often mistaken for `car`. `Bicycle` and `motorcycle` are sometimes misclassified as `car`, presumably because they are sometimes spatially close (within the 2m match threshold) to `cars`. `Adults` have similar appearance to `police-officer` and `construction-worker`, and they are often co-localized with `child` and `stroller`; all of these might cause significant class confusion.

that classes are most often misclassified as their LCA=1 siblings within coarse-grained superclasses. We use confusion matrices to further analyze the misclassifications within superclasses, as shown in Fig. 4. Below, we explain how to compute a confusion matrix for the detection task.

For each superclass, we make a confusion matrix, in which the entry $(i, j)$ indicates the misclassification rate of class-$i$ objects as class-$j$. Specifically, given a fine-grained class $i$, we find its predictions that match ground-truth boxes within 2m center-distance of class-$i$ and all its sibling classes (LCA=1, within the corresponding superclass); we ignore all unmatched detections. This allows us to count the mis-classifications of class-$i$ objects into class-$j$, with which a simple normalization produces misclassification rates.

**Limitations.** LT3D emphasizes object detection for `rare` classes which can be safety-critical for downstream AV tasks such as motion planning and collision avoidance. However, our work does not study how solving LT3D directly affects these tasks. Future work should address this limitation. Another limitation, shared by contemporary benchmarks, is that our setup does not consider the correlation between individual classes. For example, the rare-class `stroller` is often pushed by an `adult`. One may argue that detecting `adult` is sufficient for safe navigation. However, edge cases can occur in the real world where a `stroller` can be unattended.

## 6 Conclusion

We explore the problem of long-tailed 3D detection (LT3D), detecting objects not only from `common` classes but also from many `rare` classes. This problem is motivated by the operational safety of autonomous vehicles (AVs), but has broad robotic applications, e.g., elder-assistive robots [52] that fetch diverse items [53] should address LT3D. To study LT3D, we establish rigorous evaluation protocols that allow for partial credit to better diagnose 3D detectors. We propose several algorithmic innovations to improve LT3D, including a group-free detector head, hierarchical losses that promote feature sharing across long-tailed classes, and a simple multimodal fusion method that uses RGB-based detections to filter LiDAR-based detections, achieving significant improvement for LT3D.

## Acknowledgement

This work was supported by the CMU Argo AI Center for Autonomous Vehicle Research.

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
