# OpenReview forum: "Towards Long-Tailed 3D Detection"
_robot-learning.org/CoRL/2022/Conference — CoRL 2022 Poster_

### Official Review · Reviewer_4cQJ · 2022-07-30

**Originality:** Very Good
**Technical Quality:** Very Good
**Clarity Of Presentation:** Very Good
**Impact:** 4

**Recommendation:**

Weak Accept: I recommend accepting the paper, but will not argue for my recommendation if the majority of other reviewers have a different opinion.

**Summary:**

This paper studies a very important problem in autonomous driving: long-tail 3D object detection. The authors propose several useful training methods and did an in-depth evaluation, which gives the readers some valuable insights.

**Issues:**

What if the lidar detector gives many false positives along a single line of sight on the RGB image? How can the RGB-based detector classifiy which one is the most correct?

**Quality Of The Limitations Section:**

Limitations are addressed clearly

**Reviewer Expertise:**

4: The reviewer is confident but not absolutely certain that the evaluation is correct

**Robotics Focus:**

Highly relevant to robotics but no hardware experiments

**Strengths And Weaknesses:**

### Strengths:

1. The in-depth analysis in both the method and experiment section provides the readers some very useful insights on training 3D object detectors in long-tail classes, such as using group-free detection heads and training with semantics hierarchies without softmax classification.

2. The question “Are objects from rare classes sufficiently localized but mis-classified” is very valuable. The authors claimed and showed that the LiDAR detector can reach a high recall in terms of 3D localization, but poor at classification. Then the authors use RGB modality to filter out the false positives, and improve the classification performance.

### Weaknesses:

1. The paper does not propose a significantly novel framework. But the analysis and insights are interesting.

In summary, the paper provides a valuable analysis on training 3D object detectors for long-tail classes. I recommend for publication because this paper could potentially inspire new methods.


**Summary Of Recommendation:**

Recommend for publication

---

> ### Author Response · Authors · 2022-08-26
> **Response to Reviewer 4cQJ**
>
> We thank Reviewer 4cQJ for the insightful comment and positive rating (Weak Accept)! You have a wonderful point below.
>
> >**Reviewer 4cQJ asks how the RGB-based detector would classify the most correct LiDAR detection if the lidar detector gives many false positives along a single line of the RGB image.**
>
> Great question! In this scenario, our method Multi-Modal Fusion by Filtering (Line 139) will first place the RBG-based 3D detection in the BEV (Fig.2), keep the highest-scoring LiDAR-detection within a predefined radius centered at the RGB-detection, and remove the remaining LiDAR-detections along the line of sight.

---

### Official Review · Reviewer_bNMv · 2022-07-31

**Originality:** Fair
**Technical Quality:** Good
**Clarity Of Presentation:** Excellent
**Impact:** 3

**Recommendation:**

Weak Reject: I recommend rejecting the paper, but will not argue for my recommendation if the majority of other reviewers have a different opinion.

**Summary:**

This paper studies the Long-Tailed 3D Detection (LT3D) problem in the application domain of autonomous vehicles (AVs). It argues that accurately detecting long-tailed classes in AV datasets is crucial to their safe and effective deployment in real-world applications. Most current benchmarks, however, either: (a) do not measure the performance of state-of-the-art detectors on these long-tailed classes, or (b) group them with other, more commonly occurring classes, inhibiting detector performance on these classes. The paper addresses these concerns by proposing architectural improvements, such as training with semantic hierarchies and multi-modal filtering. It also presents a hierarchical evaluation metric for measuring the performance of a detector on all classes, including long-tailed classes. The experimental evaluation evaluates different detectors on the nuScenes dataset according to the proposed metric.

**Issues:**

Better support the novelty and significance of the work in context of prior and ongoing work in the area of computer vision algorithms focusing on long-tailed detection.

Experiments on additional datasets. The semantic hierarchy can be manually handcrafted. It would be acceptable to evaluate only the CenterPoint and Transfusion detectors.


**Quality Of The Limitations Section:**

Additional details required

**Reviewer Expertise:**

3: The reviewer is fairly confident that the evaluation is correct

**Robotics Focus:**

Highly relevant to robotics but no hardware experiments

**Strengths And Weaknesses:**

On the positive side, the paper is well-written and the problem of LT3D is clearly defined. The experimental section is thorough in terms of the detection methods being evaluated. It evaluates both multi-modal and non-multi-modal detectors, which is appreciated. It also presents an ablation study of the different detector components proposed in Section 3. Table 1 summarizes the state-of-the-art in this problem domain and highlight the key insights from different detection methods. It presents clearly the technical improvements proposed to state-of-the-art detectors and the evaluation metrics designed to measure LT3D performance.  The survey of related work also appears to be representative of some of the related work.

In terms of limitations, the main concern is that there has been a lot of work in long-tailed recognition, especially for the 2D case, with similar ideas to those proposed in the current paper on how to address this challenge (e.g., hierarchical reasoning). Some of these efforts are mentioned in Section 2. The paper does not do a good job at this point, however, of highlighting how the current work differentiates from them beyond trying the same ideas in the 3D domain.  Stronger justification beyond “...rare classes are not only infrequent, but are also difficult to distinguish using LiDAR alone” would be helpful to emphasize the importance of the challenge in this domain. Because the proposed solution appears to be rather standard given the literature and it is not clear how the nature of the LiDAR data affects the approach.

While there is some improvement in detection using hierarchies and multimodal filtering (Table 2), it does not reach the point of sufficiently supporting the claim that these solutions are addressing the challenge. For instance, the mAP_H values for the "Child" and "Pushable-Pullable" classes are still rather low.

Furthermore, the experimental evaluation only considers a single dataset, nuScenes. The argument made in Section 5.1 is that nuScenes explicitly defines the semantic hierarchies required for the evaluation protocol proposed in the paper. Nevertheless, it should be possible to use external semantic hierarchies (e.g., WordNet) for other dataset labels.

In Section 3, Line 110, it is stated that "...heads of rare classes overfit..". Then, in Section 3, Line 120, "...use a per-class logit-loss rather than multi-class softmax loss...". Why does the latter not suffer from the same overfitting problem for rare classes.


**Summary Of Recommendation:**

The paper is well written and the technical direction for approaching the problem of LT3D is sound, while there are accompanying experiments that use state-of-the-art detectors. The novelty, however, is not clear given the amount of work in long-tailed detection and the manuscript does not sufficiently differentiate the challenge from the work in 2D detection. The arguments would also be better supported if the proposed changes were evaluated on more than a single dataset, where it may be possible to argue stronger improvements.

---

> ### Author Response · Authors · 2022-08-26
> **Response to Reviewer bNMv (1/2)**
>
> We thank Reviewer bNMv for the insightful comments! Our response to their questions and comments follows.
>
> >**Reviewer bNMv is concerned that our work does not sufficiently differentiate from 2D long-tailed recognition.**
>
> Other reviewers (AC UhBN, Reviewer LdEV and Reviewer 4cQJ) see the extension of long-tailed recognition from 2D to 3D-detection as a valuable step. That said, we agree that can better explain the differences. Compared to 2D image-based recognition, 3D long-tailed detection has unique opportunities and challenges because sensors such as LIDAR directly provide geometric and ego-motion cues that are difficult to extract from 2D images. 2D detectors must detect objects of different scales due to perspective image projection, dramatically increasing the complexity of the output space (e.g., requiring more anchor boxes). In contrast, 3D objects do not exhibit as much scale variation, but far-away objects tend to be more sparse, imposing different challenges. Finally, 3D detectors often use class-aware heads (i.e. each class has its own binary classifier) while 2D long-tail recognition typically use shared softmax heads (which may make it easier to enforce hierarchies, as explained above).  Our work shows that leveraging both LIDAR and RGB modalities boosts long-tailed 3D detection, encouraging the robotics community to revisit multimodal fusion in the context of LT3D.
>
> >**Reviewer bNMv is concerned that we use only one dataset (nuScenes) in the experiment and questions whether our results generalize to other datasets.**
>
> Thank you for pushing us to finish new experiments on another long-tailed benchmark Argoverse 2.0 (AV2) [1]. As AV2 does not define a semantic hierarchy, we adapt the nuScenes hierarchy (recall that you suggested using some defined hierarchies such as by WordNet) for AV2.
>
> Clearly, our main conclusions still hold. Compared to the CenterPoint reported [1], our implementation improves mAP by 14.4 by carefully adopting LiDAR sweep aggregation and class-balanced sampling. Based on our implementation, exploiting hierarchical semantics improves mAP from 35.8 to 38.4. Note that Argoverse 2.0 does not provide a semantic hierarchy; we construct one (see table below) based on the nuScenes hierarchy (Fig. 3). Interestingly, FCOS3D performs significantly worse than the LiDAR based detectors, yet multimodal filtering improves mAP to 48.4 mAP. These new results on Argoverse 2.0 are consistent with those on nuScenes, demonstrating the general applicability of our approach.
> |                   | Objects of Interest |                      |
> |:-----------------:|:-------------------:|:--------------------:|
> |      Vehicle      |      Vulnerable     |        Movable       |
> |  Regular Vehicle  |      Pedestrian     |        Bollard       |
> |   Large Vehicle   |    Wheeled Rider    |   Construction Cone  |
> |        Bus        |       Biyclist      |         Sign         |
> |     Box Truck     |     Motorcylcist    |  Construction Barrel |
> |       Truck       |       Bicycle       |       Stop Sign      |
> | Vehicular Trailer |      Motorcycle     | Mobile Crossing Sign |
> |     Truck Cab     |    Wheeled Device   |     Message Board    |
> |     School Bus    |       Stroller      |                      |
> |  Articulated Bus  |         Dog         |                      |
>
>
> | Method                                          | Multimodal   | Many | Medium | Few  | All  |
> |-------------------------------------------------|--------------|------|--------|------|------|
> | FCOS3D                                          |              | 27.2 | 17.0   | 7.8  | 14.6 |
> | CenterPoint (LiDAR-only) reported by [1]       |              | 66.7 | 32.9   | 14.1 | 29.6 |
> | CenterPoint (LiDAR-only) [our reimplementation] |              | 77.4 | 46.9   | 30.2 | 44.0 |
> | CenterPOint (LiDAR-only) + Hierarchy            |              | 79.0 | 50.3   | 33.6 | 47.0 |
> |                          + Filtering            | $\checkmark$ | 79.0 | 51.4   | 35.3 | 48.4 |

---

> > ### Author Response · Authors · 2022-08-26
> > **Response to Reviewer bNMv (2/2)**
> >
> > >**Reviewer bNMv comments that the numbers for rare classes such as child are still low w.r.t mAP_H, and this does not sufficiently support the claim that the solutions are addressing the challenge of LT3D.**
> >
> > Reviewer bNMv is correct that the numbers are still low for some rare-classes such as child, manifesting the difficulty of LT3D. It is worth noting that our solutions boost the performance (in mAP_H) on child from 0.1 to 5.3 with CenterPoint (cf. Table 2 on LCA=0); if allowing for partial credits with LCA=2 (e.g., allowing child to be misclassified as adult), the number increases to 16.9 mAP_H! Compared to car (on which our solutions increase from mAP_H=86.5 to 88.5 with LCA=0 and 89.5 with LCA=2), the improvement on child is significant and encouraging. We believe our solutions (including metrics and methods) are insightful to the community that will spur broad discussions.
> >
> > >**Reviewer bNMv asks how our group-free architecture prevents overfitting for rare classes.**
> >
> > To answer your question, let's contrast the multi-head and single-head (i.e. group-free) architectures. The multi-head architecture has per-class detectors that consist of multiple layers with lots of parameters, hence learning them easily overfits to rare-classes. In contrast, group-free architecture shares backbone across all classes, and each class has only one linear layer as its detector. This significantly reduces the number of parameters and allows learning the shared feature backbone collaboratively with all classes, effectively mitigating overfitting to rare-classes.
> >
> > [1] Wilson et al. "Argoverse 2: Next Generation Datasets for Self-Driving Perception and Forecasting." NeurIPS Datasets and Benchmarks Track. 2021.

---

### Official Review · Reviewer_LdEV · 2022-08-01

**Originality:** Very Good
**Technical Quality:** Good
**Clarity Of Presentation:** Excellent
**Impact:** 4

**Recommendation:**

Weak Accept: I recommend accepting the paper, but will not argue for my recommendation if the majority of other reviewers have a different opinion.

**Summary:**

This paper studies the problem of 3D detection and classification of rare objects in urban driving. It proposes to use hierarchical classification to detect rare objects even if they may not be classified correctly. The authors develop a method to train hierarchical 3D object detectors from either RGB or LIDAR and an evaluation protocol to measure detection performance under hierarchies. Evaluation is performed on the real-world nuScenes dataset, but no robot demonstration or validation in deployment is offered.

**Issues:**

Most importantly, I recommend the authors to address the points listed under 'weaknesses'.

Additionally, here are more detailed notes of issues I found in the submission:

- The abstract claims that you would propose "hierarchical losses". The losses that you use are not described, but from context it seems to me that these losses actually do not enforce any hierarchy, but just enable a detection to have multiple class-assignments (that do not necessarily have to have a hierarchy).
- You claim to boost performance in the problem of 3D object detection (line 55). I think this claim goes too far, as it implies that your training is better than the established approaches on nuscenes. For this, a comparison to the standard training (with the coarser classes) would be necessary. It would actually be interesting to see if the added information of more fine-grained classes boosts performance compared to training with only coarse classes. For the current state of the paper, a formulation like "compared to naive training on the long-trailed class distribution" is necessary in line 55.
- line 73: typo (RGB)
- line 122: Here, please explain in detail which loss you use and why. How does it relate to existing works on hierarchical classification?
- line 122: Personally, I find 'cross-entropy loss' more clear than 'softmax loss'.
- line 127: This is pure speculation. Please remove or provide an argument.
- line 168: I recommend that you explain more clearly how your proposed metric relates to [43]. Is it a similar methodology of LCA levels, just applied to a different base metric? something entirely different?
- 19 classes does not seem particularly long-tailed compared to e.g. class distributions in Mapillary Vistas. I understand the argument that the rare classes are tail-classes because they are orders of magnitude less frequent than the majority classes. However, I think it would help to quantify and relate this: How many orders of magnitude are between tail classes and majority classes in other datasets? Are results from the nuScene dataset with only 19 classes relevant to long-tailed classification in general?
- lines 216ff: Please discuss your training approach in relation to other works. Why did you follow this approach? How does it relate to existing hierarchical classification approaches? What is novel about your approach?
- line 216: Do you also modify the RGB-based detectors?
- Table 1: Is '+Camera' for TransFusion a typo? If not, what does it mean? Do you not train TransFusion with hierarchies?
- lines 240-241: Why did you downsample instead of using crops during training? Does a single image for inference not fit into memory? Also, please indicate in all tables that TransFusion has less information available (downsampling makes detection of far-away objects harder) compared to your multimodal filtering approach.
- Table 2: Can you please add here a comparison with a detector just trained on the coarse training protocol from nuScenes (basically trained directly only on the LCA=1 level)? The same could be done with LCA=2. Both would give a good indication of what established training methods would achieve for this evaluation level, and give context what a potential 'upper bound' is.


**Quality Of The Limitations Section:**

Limitations are addressed clearly

**Reviewer Expertise:**

4: The reviewer is confident but not absolutely certain that the evaluation is correct

**Robotics Focus:**

Relevant but unlikely to deploy to hardware in near future

**Strengths And Weaknesses:**

**strengths**

- Hierarchical classification and long-tail detection are underexplored and relevant problems in robotic perception, beyond the studied application of autonomous driving.
- The proposed methodology is evaluated across a range of different architectures and methods, showing a general effectiveness.
- The paper is very well written and the detailed evaluation protocol is of particular value to spark future work in this area.
- The multimodal filtering trick is simple yet effective, even when applied to methods that are already trained to fuse modalities.

**weaknesses**

- No improvement over existing methods for hierarchical classification. It seems that actually most of the tested existing methods have better long-tail performance than the proposed method.
- There are no robotic experiments. This is a computer-vision paper, but with high relevance to robotic perception.

**points resolved through the revision**
- The authors do not describe how their models are trained. They describe in general that they do not use a 'softmax loss' (I assume they mean cross-entropy), but don't describe what they use instead. They also do not describe how the RGB methods are altered for hierarchical classification.
- There is no comparison to related work in hierarchical classification. It is unclear whether the hierarchical training is novel or useful compared to existing methods for training hierarchical classification (that exist, according to the related work section).
- Given the lack of implementation detail in the manuscript and no code supplement, the method and the results are not reproducible. (The authors claim to open-source their code later on, but for CoRL authors should use the offered option of the code supplement or describe their method better in the text.)

**Summary Of Recommendation:**

**summary before revision:**
I think this paper pushes into the right direction, emphasising a number of problems with current detection methods (not only in 3D) and coming up with sensible ideas. It is valuable to discuss these ideas and problems at CoRL. The biggest weakness of this paper is that it is unclear to me how much of the proposed method is original and how much adapted from problems like image classification. Further, the results are not reproducible due to lack of details. Both issues however can be rather easily addressed through discussion with the authors and revision.

**summary after revision:**
I think this paper pushes into the right direction, emphasising a number of problems with current detection methods (not only in 3D) and coming up with sensible ideas. It is valuable to discuss these ideas and problems at CoRL. The comparison with existing hierarchical classification works however shows no improvement of this particular method. Existing methods even seem to perform better in the long-tail classes. There are still many valuable ideas and approaches in this paper, but this comparison makes the contribution less strong than I initially thought.

---

> ### Author Response · Authors · 2022-08-26
> **Response to Reviewer LdEV (1/5)**
>
> We thank Reviewer LdEV for the detailed and constructive comments, and the positive rating (Weak Accept)! We will revise minor issues such as typos; we address your major questions below.
>
> > **Reviewer LdEV asks how the models are trained.**
>
> We apologize for the lack of training details. In short, we follow the training procedure of the respective detectors [1,2,3,4,5] which have open-source code [9,10,11,12]. We describe important implementation details below.
> - As for *model architecture*, we adopt the architecture in [2] but make an important modification. The original architecture (for the standard nuScenes benchmark) has six heads designed for ten classes; each head has 64 filters. We first adapted this architecture for LT3D using seven heads designed for 19 classes, termed multi-head. We then replace these seven heads with a single head consisting of 512 filters shared by all classes. We find this modification reduces memory use and maintains state-of-the-art performance as shown in Table 1 (copied below) for the original nuScenes benchmark. Our proposed single-head detector architecture outperforms the multi-head CenterPoint on LT3D. Note, TC is traffic-cone, CV is construction-vehicle, MC is motorcycle, PP is pushable-pullable, CW is construction-worker, and PO is police-officer. We highlight Medium and Few classes in blue (which were ignored in the original nuScenes benchmark).
> | **CenterPoint** | **Multi-Head** | **Car**   | **Ped.** | **Barrier** | **TC**     | **Truck** | **Bus**      | **Trailer** | **CV**   | **MC**   | **Bicycle** |
> |-----------------|----------------|-----------|----------|-------------|------------|-----------|--------------|-------------|----------|----------|-------------|
> | **Original**    | $\checkmark$   | 87.7      | 87.7     | 70.7        | 74.0       | 63.6      | 72.7         | **45.1**    | **26.3** | 64.7     | 47.9        |
> |                 |                | **89.1**  | **88.4** | **70.8**    | **74.3**   | **64.8**  | **72.9**     | 42.0        | 25.7     | **65.9** | **53.6**    |
> | **for LT3D**    | $\checkmark$   | 82.4      | -        | 62.0        | 60.1       | 49.4      | 55.7         | 28.9        | 9.7      | 48.9     | 33.6        |
> |                 |                | **88.1**  | -        | **72.4**    | **72.7**   | **62.7**  | **70.8**     | **40.2**    | **24.5** | **62.8** | **48.5**    |
> |                 |                | **Adult** | **PP**   | **CW**      | **Debris** | **Child** | **Stroller** | **PO**      | **EV**   | **PM**   | **All**     |
> | **Original**    | $\checkmark$   | -         | -        | -           | -          | -         | -            | -           | -        | -        | 64.0        |
> |                 |                | -         | -        | -           | -          | -         | -            | -           | -        | -        | **64.8**    |
> | **for LT3D**    | $\checkmark$   | 81.2      | 21.7     | 14.2        | 1.1        | **0.1**   | 0.1          | 1.3         | 0.1      | **0.1**  | 31.2        |
> |                 |                | **86.3**  | **32.7** | **22.2**    | **4.3**    | **0.1**   | **4.3**      | **1.8**     | **10.3** | **0.1**  | **39.2**    |
> - As for *training losses*, we use the sigmoid focal loss (for recognition) [8] and L1 regression loss (for localization) below. Existing works also use the same losses but only with fine labels; we apply the loss to both coarse and fine labels. Concretely, our loss function for CenterPoint is as follows: $L = L_{HM} + \lambda L_{REG}$, where $L_{HM} = \sum_{i=0}^{C} SigmoidFocalLoss(X_i, Y_i)$ and $L_{REG} = |X_{BOX} - Y_{BOX}|$, where $X_i$ and $Y_i$ are the $i^{th}$ class' predicted and ground-truth heat maps, while $X_{BOX}$ and $Y_{BOX}$ are the predicted and ground-truth box attributes. Without our hierarchical loss, C=19. With our hierarchical loss, C=23 (19 Fine Grained + 3 Coarse + 1 Object Class). $\lambda$ is set to 0.25.

---

> > ### Author Response · Authors · 2022-08-26
> > **Response to LdEV (2/5)**
> >
> > - As for *post-processing*, we use non-maximum suppression (NMS) on detections *within* each class to suppress lower-scoring detections. In contrast, existing works apply NMS on all detections *across* classes, i.e., suppressing detections overlapping other classes' detections (e.g., a pedestrian detection can suppress other pedestrian and traffic cone detections). We find that applying within-class NMS improves overall performance by 2 AP compared to across-class NMS (Table 2), but decreases performance on Few classes by 5 AP. We posit that detections for Few classes are noisy and contain many false positives, which are correctly suppressed by NMS’ing across other class detections. This inspired us to develop a simple hybrid NMS: apply within-class NMS for Many and Medium classes and across-class NMS for Few classes. This further boosts performance by 1.8 and 3.7 mAP over within-class and cross-class NMS, respectively. We sincerely thank the AC and reviewers for this comment, without which we would not have improved this far!
> > | Method           | Many | Medium | Few | All  |
> > |------------------|------|--------|-----|------|
> > | Across-class NMS | 74.2 | 43.0   | **9.5** | 38.5 |
> > | Within-class NMS | **77.1** | **45.1**   | 4.3 | 40.4 |
> > | Hybrid           | **77.1** | **45.1**   | **9.5** | **42.2** |
> >
> > >**Reviewer LdEV asks for comparisons between our hierarchical loss to prior work.**
> >
> > Good suggestion! Classic methods train a hierarchical softmax (in contrast to our simple approach of sigmoid focal loss with both fine and coarse classes), where one multiplies the class probabilities of the hierarchical predictions during training and inference [7]. We implemented such an approach, but found the training did not converge. Interestingly, [7] shows such a hierarchical softmax loss has little impact on long-tailed object detection (in 2D images), which is one reason they have not been historically adopted. Instead, we found better results using the method from [6] (a winning 2D object detection system on the LVIS benchmark) which multiples class probabilities of predictions (e.g. $P_{CAR} = P_{OBJ} * P_{CAR}) at test-time, even when such predictions are not trained with a hierarchical softmax. We tested three variants and compared it to our approach (which recall, uses only fine-grained class probabilities at inference):
> > - (a) Ours (Lines 115 - 127)
> > - (b) ObjectScore * FinegrainScore [6], e.g. $P_{CAR} = P_{OBJ} * P_{CAR}$
> > - (c) Variant 1 of [6], i.e. CoarseScore * FinegrainScore, e.g. $P_{CAR} = P_{VEHICLE} * P_{CAR} $
> > - (d) Variant 2 of [6], i.e. ObjectScore * CoarseScore * FinegrainScore, e.g. $P_{CAR} = P_{OBJ} * P_{VEHICLE} * P_{CAR} $
> >
> >  Different variants achieve similar performance. We note that other methods do improve accuracy in the tail by sacrificing performance in the head, suggesting that hybrid approaches that apply different techniques for head-vs-tail classes may further improve accuracy. Unlike [6,7] which requires a strict label hierarchy, our approach is not limited to a hierarchy (as Reviewer LdEV points out!).
> > | Method                     | Many | Medium | Few | All  |
> > |----------------------------|------|--------|-----|------|
> > | CenterPoint                | 73.7 | 41.3   | 3.0 | 37.5 |
> > | (a) Ours                   | **77.1** | 45.1   | 4.3 | 40.4 |
> > | (b) Object * FineGrain [6] | 76.4 | 45.0   | 5.3 | 40.5 |
> > | (c) Variant-1 of [6]       | 76.5 | **45.2**   | 5.2 | **40.6**|
> > | (d) Variant-2 of [6]       | 74.5 | 43.5   | **5.6** | 39.5 |
> >
> > >**Reviewer LdEV comments that the code is not provided and it is unclear how to reproduce the results.**
> >
> > We did not attach code along with our submission because we find it takes time to make an accessible version to the community. Our codebase modified four different open-source repositories ([9], [10], [11], [12]). In principle, one can readily reproduce the results by repurposing these open-sourced repositories, particularly for the simple methods introduced in our paper (exploiting semantic hierarchies in Line 115 and multimodal filtering in Line 134). Because of its simplicity, Reviewer b6Vx finds “the techniques introduced in the paper are simple to implement”. Yet, we agree that diligent work should be done to guarantee an accessible open-source codebase for the community, and commit to releasing our cleaned-up code later to foster LT3D research.
> >
> > >**Reviewer LdEV points out that the proposed hierarchical loss does not enforce a hierarchy.**
> >
> > Correct! The loss simply trains coarse-grained classification heads, which do not necessarily enforce any hierarchy. Empirical results show that this loss helps learn better models that improves detection performance (cf. Table 1).

---

> > > ### Author Response · Authors · 2022-08-26
> > > **Response to LdEV (3/5)**
> > >
> > > >**Reviewer LdEV thinks the claim "the method boosts performance in the problem of 3D object detection" goes too far.**
> > >
> > > Good catch! We meant "the method boosts performance in the problem of long-tailed 3D detection (LT3D)", not the standard 3D object detection task. We will revise in the camera-ready.
> > >
> > > >**Reviewer LdEV asks what losses are used that make it "trivial" to add additional coarse classes in Line 122.**
> > >
> > > We use per-class sigmoid focal loss, meaning that each class has its own predictor modeled as a linear layer over the shared backbone. This makes it trivial to add additional coarse classes without changing the backbone architecture. We will clarify.
> > >
> > > >**Reviewer LdEV asks how the proposed metric (L168) relates to [43] (i.e., [13] below).**
> > >
> > > Before addressing your confusion, we realize that we cited the wrong version of the paper [13], which does not discuss the hierarchical metric but its extended journal version [14] does. Our metric is related to [14] as both exploit semantic hierarchies in the metric to allow for partial credit from misclassifications across classes. However, our metric has two major differences. First, our metric is adapted for long-tailed 3D detection, which is non-trivial as it must consider both classification errors and localization errors (Line 170). Second, our metric studies performance for each of the levels of the least-common-ancestor (LCA) independently (Line 173-183). In contrast, [14] uses the LCA as weights in a weighted-sum of accuracies towards a single summary number.
> > >
> > > >**Reviewer LdEV suggests removing the sentence "this is presumably because it regularizes the learned features to generalize better" because it is not well-grounded (Line 127).**
> > >
> > > Agreed! We will delete this line.

---

> > > > ### Author Response · Authors · 2022-08-26
> > > > **Response to LdEV (4/5)**
> > > >
> > > > >**Reviewer LdEV suggests quantifying the imbalance between common and tail classes in nuScenes and other datasets, and asks whether the results from the nuScene dataset are relevant to long-tailed classification in general.**
> > > >
> > > > Good point! To measure the class imbalance, we use the imbalance factor (IF) defined as the ratio between the # annotations of the max-class and min-class [15]. The nuScenes dataset has an IF=1670, and Argoverse 2.0 [16] has an IF=2500. Their IFs are significantly more imbalanced than well-established benchmarks used in long-tail image classification, e.g., iNaturalist [17] that has IF=500 and ImageNet-LT [18] that has IF=1000. Furthermore, we add results on another large-scale dataset, Argoverse 2.0. Clearly, our main conclusions still hold. Compared to the CenterPoint reported [16], our implementation improves mAP by 14.4 by carefully adopting LiDAR sweep aggregation and class-balanced sampling. Based on our implementation, exploiting hierarchical semantics improves mAP from 35.8 to 38.4. Note that Argoverse 2.0 does not provide a semantic hierarchy; we construct one (see table below) based on the nuScenes hierarchy (Fig. 3). Interestingly, FCOS3D performs significantly worse than the LiDAR based detectors, yet multimodal filtering improves mAP to 48.4 mAP. These new results on Argoverse 2.0 are consistent with those on nuScenes, demonstrating the general applicability of our approach.
> > > >
> > > > |                   | Objects of Interest |                      |
> > > > |:-----------------:|:-------------------:|:--------------------:|
> > > > |      Vehicle      |      Vulnerable     |        Movable       |
> > > > |  Regular Vehicle  |      Pedestrian     |        Bollard       |
> > > > |   Large Vehicle   |    Wheeled Rider    |   Construction Cone  |
> > > > |        Bus        |       Biyclist      |         Sign         |
> > > > |     Box Truck     |     Motorcylcist    |  Construction Barrel |
> > > > |       Truck       |       Bicycle       |       Stop Sign      |
> > > > | Vehicular Trailer |      Motorcycle     | Mobile Crossing Sign |
> > > > |     Truck Cab     |    Wheeled Device   |     Message Board    |
> > > > |     School Bus    |       Stroller      |                      |
> > > > |  Articulated Bus  |         Dog         |                      |
> > > >
> > > >
> > > > | Method                                          | Multimodal   | Many | Medium | Few  | All  |
> > > > |-------------------------------------------------|--------------|------|--------|------|------|
> > > > | FCOS3D                                          |              | 27.2 | 17.0   | 7.8  | 14.6 |
> > > > | CenterPoint (LiDAR-only) reported by [16]       |              | 66.7 | 32.9   | 14.1 | 29.6 |
> > > > | CenterPoint (LiDAR-only) [our reimplementation] |              | 77.4 | 46.9   | 30.2 | 44.0 |
> > > > | CenterPOint (LiDAR-only) + Hierarchy            |              | 79.0 | 50.3   | 33.6 | 47.0 |
> > > > |                          + Filtering            | $\checkmark$ | 79.0 | 51.4   | 35.3 | 48.4 |
> > > >
> > > > >**Reviewer LdEV asks if we modify the RGB-based detectors.**
> > > >
> > > > No.
> > > >
> > > > >**Reviewer LdEV points out the typo "+Camera" (Table 1).**
> > > >
> > > > Thank you for the careful review! We will use the terminology from [5] and correct the first row to “TransFusion-L” (LiDAR-only) and the second row to “TransFusion” (LiDAR + RGB)
> > > >
> > > > >**Reviewer LdEV asks whether we train TransFusion with hierarchies.**
> > > >
> > > > No. We are unable to do so due to limited GPU memory.
> > > >
> > > > >**Reviewer LdEV asks why downsampling RGB images instead of cropping during training (L240-241).**
> > > >
> > > > Aiming to construct training batches of both LIDAR and RGB images in our limited GPU memory, we chose to downsample RGB images rather than cropping them, because cropping would remove large objects that are close to the camera due to perspective. We note that we can train at full RGB-image resolution using sufficiently large GPU memory, potentially improving the detection performance further.

---

> > > > > ### Author Response · Authors · 2022-08-26
> > > > > **Response to LdEV (5/5)**
> > > > >
> > > > > >**Reviewer LdEV asks how models would perform if trained on categories  at coarse (LCA=1) or object (LCA=2) level (Table 2), and wonders whether they would be "upper-bound"**
> > > > >
> > > > > Interesting idea! To study this, we trained CenterPoint using the class labels with different LCAs, i.e., merging fine-grained labels into superclasses (Fig. 3) in training. In testing, to benchmark on a fine-grained class such as child, we relabel  pedestrian predictions as child predictions (because child is a subclass of pedestrian). The table below compares these models. We list three salient observations.
> > > > > - Learning with LCA=1 significantly increases performance on child and stroller (both of which belong to pedestrians).
> > > > > - Learning with LCA=1 significantly decreases performance on bicycle, a subclass of vehicle.
> > > > > - Learning with LCA=2 slightly increases performance on child and stroller but greatly decreases performance on other categories.
> > > > >
> > > > > There is no clear trend on how to merge fine-grained classes in training, and these models cannot be thought of as "upper-bound" as they do not guarantee better numerical results. We posit that for rare-classes, it might be good to merge only relevant classes (e.g., training a stroller detector with wheelchairs as additional “augmented” training examples since they tend to have similar 3D shape and size). However, merging irrelevant classes (e.g., merging bicycle and truck) hurts performance. Therefore, we hypothesize that grouping rare-classes as a way of data augmentation should be class-dependent, as studied in [19].
> > > > >
> > > > > |        CenterPoint       |       |  Car | Truck | Bicycle | Stroller | Child |
> > > > > |:------------------------:|-------|:----:|:-----:|:-------:|:--------:|:-----:|
> > > > > | Train with LCA=0  (ours) | LCA=0 | 86.5 |  53.9 |   47.2  |    3.6   |  0.1  |
> > > > > |                          | LCA=1 | 87.3 |  59.5 |   48.8  |    3.8   |  0.1  |
> > > > > |                          | LCA=2 | 87.3 |  59.6 |   49.5  |    4.0   |  0.1  |
> > > > > |     Train with LCA=1     | LCA=0 | 65.8 |  9.3  |   1.6   |    0.1   |  0.1  |
> > > > > |                          | LCA=1 | 85.3 |  63.1 |   32.1  |   17.2   |  12.5 |
> > > > > |                          | LCA=2 | 85.5 |  63.6 |   33.0  |   17.4   |  12.9 |
> > > > > |     Train with LCA=2     | LCA=0 | 31.5 |  4.6  |   0.1   |    0.1   |  0.1  |
> > > > > |                          | LCA=1 | 36.3 |  8.8  |   1.8   |    0.1   |  0.1  |
> > > > > |                          | LCA=2 | 65.9 |  35.1 |   13.5  |    5.9   |  3.4  |
> > > > >
> > > > > [1] "PointPilllars: Fast Encoders for Object Detection from Point Clouds." Lang et al. CVPR 2019.
> > > > >
> > > > > [2] "Class Balanced Grouping and Sampling for Point Cloud 3D Object Detection." Zhu et al. ArXiv Tech Report.
> > > > >
> > > > > [3] "Center-based 3D Object Detection and Tracking." Yin et al. CVPR 2021.
> > > > >
> > > > > [4] "Multimodal Virtual Point 3D Detection." Yin et al. NeurIPS 2021.
> > > > >
> > > > > [5] "TransFusion: Robust LiDAR-Camera Fusion for 3D Object Detection with Transformers." Bai et al. CVPR 2022.
> > > > >
> > > > > [6] "A Hierarchical Loss and its Problems when Classifying Non-Hierarchically." Wu et al. PLOS One.
> > > > >
> > > > > [7] "Overcoming Classifier Imbalance for Long-tailed Object Detection with Balanced Group Softmax." Li et al. CVPR 2020.
> > > > >
> > > > > [8] "Focal Loss for Dense Object Detection." Lin et al. ICCV 2017.
> > > > >
> > > > > [9] https://github.com/tianweiy/CenterPoint
> > > > >
> > > > > [10] https://github.com/tianweiy/MVP
> > > > >
> > > > > [11] https://github.com/XuyangBai/TransFusion
> > > > >
> > > > > [12] https://github.com/open-mmlab/mmdetection3d
> > > > >
> > > > > [13] Deng et al. "Imagenet: A large-scale hierarchical image database." CVPR. 2009.
> > > > >
> > > > > [14] Olga, et al. "Imagenet large scale visual recognition challenge." IJCV. 2015.
> > > > >
> > > > > [15] Cao et al. "Learning imbalanced datasets with label-distribution-aware margin loss." NeurIPS. 2019.
> > > > >
> > > > > [16] Wilson et al. "Argoverse 2: Next Generation Datasets for Self-Driving Perception and Forecasting." NeurIPS Datasets and Benchmarks Track. 2021.
> > > > >
> > > > > [17] Van Horn et al. "The inaturalist species classification and detection dataset." CVPR. 2018.
> > > > >
> > > > > [18] Liu et al. "Large-scale long-tailed recognition in an open world." CVPR. 2019.
> > > > >
> > > > > [19] Randall Balestriero, Leon Bottou, and Yann LeCun, "The Effects of Regularization and Data Augmentation are Class Dependent." arXiv:2204.03632, 2022

---

> > ### Comment · Reviewer_LdEV · 2022-08-26
> > **Changes to Manuscript**
> >
> > Dear Authors,
> >
> > thanks a lot for the detailed answers. While I am still reading them in detail I would just like to note that it would be important to see how you adapt the manuscript given all the new explanations and results. Maybe I am just not finding it in openreview, but I assume you will upload a revised version?

---

> > > ### Author Response · Authors · 2022-08-26
> > > **Response to Reviewer LdEV [Changes to Manuscript]**
> > >
> > > Thank you Reviewer LdEV for the note. We agree that it's important to incorporate our new results and analyses in the camera-ready version. We are happy and eager to do so, but we are afraid that we will not be able to finish this within the limited author-reviewer discussion period.

---

> > > > ### Comment · Reviewer_LdEV · 2022-08-26
> > > > **Changes to Manuscript**
> > > >
> > > > Thank you for the quick response. I totally see how the time is limited, but I would also need to note that other submissions (also in the last year) uploaded revised manuscripts and the author instructions state "Authors will have an opportunity to submit a response to reviewers and update the papers during the discussion period.". The AC will probably have to decide how much we can take your answers into account without an updated manuscript.

---

> > > > > ### Author Response · Authors · 2022-08-27
> > > > > **will upload an updated version**
> > > > >
> > > > > Hi dear Reviewer LdEV,
> > > > >
> > > > > We have uploaded an updated version that combines the main paper and supplemental document. We hope this reference helps AC and reviewers make the final decision. We will still polish the paper further for camera-ready. Thanks again for the kind notice!

---

### Official Review · Reviewer_b6Vx · 2022-08-06

**Originality:** Fair
**Technical Quality:** Very Good
**Clarity Of Presentation:** Very Good
**Impact:** 2

**Recommendation:**

Weak Accept: I recommend accepting the paper, but will not argue for my recommendation if the majority of other reviewers have a different opinion.

**Summary:**

Existing methods for 3D classification of objects for autonomous vehicle benchmarks neglect rare classes, classes in the tail, and also ignore the hierarchies that exists among the labels, such as "child" and "construction-worker" are sub-classes of "pedestrian". The paper argues that multi-modal approaches, fusing RGB images with LiDAR deal better with the long-tailed 3D detection. Besides, the paper introduces a detection metric that favors mistakes that are less riskier, e.g., mistaking a child for an adult.

**Issues:**

The paper looks fine.

**Quality Of The Limitations Section:**

Additional details required

**Reviewer Expertise:**

2: The reviewer is willing to defend the evaluation, but it is quite likely that the reviewer did not understand central parts of the paper

**Robotics Focus:**

Irrelevant to robotics

**Strengths And Weaknesses:**

Strength:
* The paper introduces some techniques that seems to be useful to improve classification accuracy both on common and rare classes.
* The techniques introduced in the paper are simple to implement without adding too much computational burden.

Weakness:
* This is a very limited study which is unlikely to have robotic applications other than autonomous vehicles.

**Summary Of Recommendation:**

The paper in general seems to provide a simple method and convincing results. However, the relevance to the robotic field is questionable.

---

> ### Author Response · Authors · 2022-08-26
> **Response to Reviewer b6Vx**
>
> Thank Reviewer b6Vx for the positive rating (Weak Accept)! We acknowledge your concern and provide our response below.
>
> >**Reviewer b6Vx is concerned that this work on long-tailed 3D detection (LT3D) "is unlikely to have robotic applications other than autonomous vehicles."**
>
> We believe that LT3D has broad robotic applications. First, we point out that 3D detection is widely used in various robotics applications such as agricultural robots [1], robotic manipulation [2], autonomous vehicles [3], etc. Second, because natural data tends to follow long-tailed distributions, versatile robots (other than autonomous vehicles) will need to detect diverse objects in 3D and deal with the LT3D problem. For example, elder-assistive robots [4] that fetch diverse items [5] should address LT3D. Reviewer LdEV highlights that “hierarchical classification and long-tail detection are underexplored and relevant problems in robotic perception, beyond the studied application of autonomous driving.” We hope Reviewer b6Vx appreciates the generality of the LT3D problem in the context of broad and promising robotics applications.
>
> [1] Weiss and Biber, "Plant detection and mapping for agricultural robots using a 3D LIDAR sensor." Robotics and Autonomous Systems, 2011
>
> [2] Luo et al., "SKP: Semantic 3D Keypoint Detection for Category-Level Robotic Manipulation." IEEE RA-L, 2022
>
> [3] Geiger et al. "Vision meets robotics: The kitti dataset." IJRR, 2013
>
> [4] Savage N. "Robots rise to meet the challenge of caring for old people." Nature. 2022
>
> [5] Grauman et al., "Ego4D: Around the World in 3,000 Hours of Egocentric Video", CVPR, 2022

---

### Author Response · Authors · 2022-08-28
**Summary of Revisions**

Dear Area Chair and Reviewers,

We appreciate all your constructive feedback and positive ratings (three Weak Accepts and one Weak Reject). We provide detailed responses to all questions and upload a revised version for your reference (which combines the updated main paper and supplemental document). The reviewer-author interactions might make the rebuttal thread too long for you to quickly grasp how we address the pressing issues. For your convenience, we summarize our important responses below.

First, AC UhBN suggests (1) providing implementation details and (2) comparing against other hierarchical learning methods. The updated version now contains the details (mainly in Appendix A) and comparison (mainly in Appendix F).

Second, both Reviewer LdEV and bNMv (who gives Weak Reject) are interested to see results on other datasets for Long-Tailed 3D Detection (LT3D). We think it is important to run experiments on another dataset. Therefore, we benchmark methods on the Argoverse 2.0 dataset (which is long-tailed, large-scale for autonomous vehicle research). Complete results are in Appendix H. Our conclusions still hold: using hierarchical loss and multimodal filtering significantly boosts LT3D performance.

Third, Reviewer bNMv, who gives Weak Reject, finds our paper insufficiently demonstrates the differences between LT3D and 2D long-tailed recognition. Our updated version discusses their differences in Lines 93-104.

We would like to polish our paper further for the camera-ready. We hope the above summary will be helpful in your decision-making. Thank you!

---

### Meta-Review · Area_Chair_UhBN · 2022-08-13

**Recommendation:** Accept (Poster)
**Confidence:** 4

**Metareview:**

The reviewers all appreciate the relevance of the long-tailed 3D object detection, and the proposed evaluation metric used to also account for rare classes is very useful. Also, the proposed ideas of multi-modal filtering and hierarchical classification are very helpful to improve the classification performance on rare classes. On the downside, the reviewers argue that there are too little implementation details given in the paper (e.g. what is the used loss function?), and an experimental comparison to other hierarchical classification methods (which are given in the related works) is also missing. Some further discussion on these points in particular would therefore be very helpful during the rebuttal.

Post-rebuttal:
The authors have given detailed answers to most of the points raised by the reviewers. The additional experiments and evaluations give a more detailed insight into the performance of the proposed method, in particular the comparison to other hierarchy-based techniques. It would be good to move at least part of this discussion from the supplementary into the main text. Also, in Table 2 highlighting the best numbers in bold would help readability.

**Best Paper Nomination:**

No

---

> ### Author Response · Authors · 2022-08-26
> **Response to Area Chair UhBN (1/2)**
>
> We thank AC UhBN for summarizing our paper’s strengths and two issues to improve our paper. We are delighted to receive positive reviews (three Weak Accepts and one Weak Reject). We address the two issues below and also respond to reviewers separately to address their other questions.
>
> > **Details of how to train models are not sufficiently presented. The paper should describe the training approach in relation to other works**
>
> We apologize for the lack of training details. In short, we follow the training procedure of the respective detectors [1,2,3,4,5] which have open-source code [9,10,11,12]. We describe important implementation details below.
> - As for *model architecture*, we adopt the architecture in [2] but make an important modification. The original architecture (for the standard nuScenes benchmark) has six heads designed for ten classes; each head has 64 filters. We first adapted this architecture for LT3D using seven heads designed for 19 classes, termed multi-head. We then replace these seven heads with a single head consisting of 512 filters shared by all classes. We find this modification reduces memory use and maintains state-of-the-art performance as shown in Table 1 (copied below) for the original nuScenes benchmark. Our proposed single-head detector architecture outperforms the multi-head CenterPoint on LT3D. Note, TC is traffic-cone, CV is construction-vehicle, MC is motorcycle, PP is pushable-pullable, CW is construction-worker, and PO is police-officer. We highlight Medium and Few classes in blue (which were ignored in the original nuScenes benchmark).
> | **CenterPoint** | **Multi-Head** | **Car**   | **Ped.** | **Barrier** | **TC**     | **Truck** | **Bus**      | **Trailer** | **CV**   | **MC**   | **Bicycle** |
> |-----------------|----------------|-----------|----------|-------------|------------|-----------|--------------|-------------|----------|----------|-------------|
> | **Original**    | $\checkmark$   | 87.7      | 87.7     | 70.7        | 74.0       | 63.6      | 72.7         | **45.1**    | **26.3** | 64.7     | 47.9        |
> |                 |                | **89.1**  | **88.4** | **70.8**    | **74.3**   | **64.8**  | **72.9**     | 42.0        | 25.7     | **65.9** | **53.6**    |
> | **for LT3D**    | $\checkmark$   | 82.4      | -        | 62.0        | 60.1       | 49.4      | 55.7         | 28.9        | 9.7      | 48.9     | 33.6        |
> |                 |                | **88.1**  | -        | **72.4**    | **72.7**   | **62.7**  | **70.8**     | **40.2**    | **24.5** | **62.8** | **48.5**    |
> |                 |                | **Adult** | **PP**   | **CW**      | **Debris** | **Child** | **Stroller** | **PO**      | **EV**   | **PM**   | **All**     |
> | **Original**    | $\checkmark$   | -         | -        | -           | -          | -         | -            | -           | -        | -        | 64.0        |
> |                 |                | -         | -        | -           | -          | -         | -            | -           | -        | -        | **64.8**    |
> | **for LT3D**    | $\checkmark$   | 81.2      | 21.7     | 14.2        | 1.1        | **0.1**   | 0.1          | 1.3         | 0.1      | **0.1**  | 31.2        |
> |                 |                | **86.3**  | **32.7** | **22.2**    | **4.3**    | **0.1**   | **4.3**      | **1.8**     | **10.3** | **0.1**  | **39.2**    |
> - As for *training losses*, we use the sigmoid focal loss (for recognition) [8] and L1 regression loss (for localization) below. Existing works also use the same losses but only with fine labels; we apply the loss to both coarse and fine labels. Concretely, our loss function for CenterPoint is as follows: $L = L_{HM} + \lambda L_{REG}$, where $L_{HM} = \sum_{i=0}^{C} SigmoidFocalLoss(X_i, Y_i)$ and $L_{REG} = |X_{BOX} - Y_{BOX}|$, where $X_i$ and $Y_i$ are the $i^{th}$ class' predicted and ground-truth heat maps, while $X_{BOX}$ and $Y_{BOX}$ are the predicted and ground-truth box attributes. Without our hierarchical loss, C=19. With our hierarchical loss, C=23 (19 Fine Grained + 3 Coarse + 1 Object Class). $\lambda$ is set to 0.25.

---

> > ### Author Response · Authors · 2022-08-26
> > **Response to Area Chair UhBN  (2/2 continued)**
> >
> > - As for *post-processing*, we use non-maximum suppression (NMS) on detections *within* each class to suppress lower-scoring detections. In contrast, existing works apply NMS on all detections *across* classes, i.e., suppressing detections overlapping other classes' detections (e.g., a pedestrian detection can suppress other pedestrian and traffic cone detections). We find that applying within-class NMS improves overall performance by 2 AP compared to across-class NMS (Table 2), but decreases performance on Few classes by 5 AP. We posit that detections for Few classes are noisy and contain many false positives, which are correctly suppressed by NMS’ing across other class detections. This inspired us to develop a simple hybrid NMS: apply within-class NMS for Many and Medium classes and across-class NMS for Few classes. This further boosts performance by 1.8 and 3.7 mAP over within-class and cross-class NMS, respectively. We sincerely thank the AC and reviewers for this comment, without which we would not have improved this far!
> > | Method           | Many | Medium | Few | All  |
> > |------------------|------|--------|-----|------|
> > | Across-class NMS | 74.2 | 43.0   | **9.5** | 38.5 |
> > | Within-class NMS | **77.1** | **45.1**   | 4.3 | 40.4 |
> > | Hybrid           | **77.1** | **45.1**   | **9.5** | **42.2** |
> >
> > > **The paper needs to be compared with related work in hierarchical classification**
> >
> > Good suggestion! Classic methods train a hierarchical softmax (in contrast to our simple approach of sigmoid focal loss with both fine and coarse classes), where one multiplies the class probabilities of the hierarchical predictions during training and inference [7]. We implemented such an approach, but found the training did not converge. Interestingly, [7] shows such a hierarchical softmax loss has little impact on long-tailed object detection (in 2D images), which is one reason they have not been historically adopted. Instead, we found better results using the method from [6] (a winning 2D object detection system on the LVIS benchmark) which multiples class probabilities of predictions (e.g. $P_{CAR} = P_{OBJ} * P_{CAR}) at test-time, even when such predictions are not trained with a hierarchical softmax. We tested three variants and compared it to our approach (which recall, uses only fine-grained class probabilities at inference):
> > - (a) Ours (Lines 115 - 127)
> > - (b) ObjectScore * FinegrainScore [6], e.g. $P_{CAR} = P_{OBJ} * P_{CAR}$
> > - (c) Variant 1 of [6], i.e. CoarseScore * FinegrainScore, e.g. $P_{CAR} = P_{VEHICLE} * P_{CAR} $
> > - (d) Variant 2 of [6], i.e. ObjectScore * CoarseScore * FinegrainScore, e.g. $P_{CAR} = P_{OBJ} * P_{VEHICLE} * P_{CAR} $
> >
> >  Different variants achieve similar performance. We note that other methods do improve accuracy in the tail by sacrificing performance in the head, suggesting that hybrid approaches that apply different techniques for head-vs-tail classes may further improve accuracy. Unlike [6,7] which require a strict label hierarchy, our approach is not limited to a hierarchy (as Reviewer LdEV points out!).
> > | Method                     | Many | Medium | Few | All  |
> > |----------------------------|------|--------|-----|------|
> > | CenterPoint                | 73.7 | 41.3   | 3.0 | 37.5 |
> > | (a) Ours                   | **77.1** | 45.1   | 4.3 | 40.4 |
> > | (b) Object * FineGrain [6] | 76.4 | 45.0   | 5.3 | 40.5 |
> > | (c) Variant-1 of [6]       | 76.5 | **45.2**   | 5.2 | **40.6**|
> > | (d) Variant-2 of [6]       | 74.5 | 43.5   | **5.6** | 39.5 |
> >
> > [1] "PointPilllars: Fast Encoders for Object Detection from Point Clouds." Lang et al. CVPR 2019.
> >
> > [2] "Class Balanced Grouping and Sampling for Point Cloud 3D Object Detection." Zhu et al. ArXiv Tech Report.
> >
> > [3] "Center-based 3D Object Detection and Tracking." Yin et al. CVPR 2021.
> >
> > [4] "Multimodal Virtual Point 3D Detection." Yin et al. NeurIPS 2021.
> >
> > [5] "TransFusion: Robust LiDAR-Camera Fusion for 3D Object Detection with Transformers." Bai et al. CVPR 2022.
> >
> > [6] "A Hierarchical Loss and its Problems when Classifying Non-Hierarchically." Wu et al. PLOS One.
> >
> > [7] "Overcoming Classifier Imbalance for Long-tailed Object Detection with Balanced Group Softmax." Li et al. CVPR 2020.
> >
> > [8] "Focal Loss for Dense Object Detection." Lin et al. ICCV 2017.
> >
> > [9] https://github.com/tianweiy/CenterPoint
> >
> > [10] https://github.com/tianweiy/MVP
> >
> > [11] https://github.com/XuyangBai/TransFusion
> >
> > [12] https://github.com/open-mmlab/mmdetection3d